# Characteristics of Systemic Lupus Erythematosus Patients with Diffuse Alveolar Hemorrhage: Clinical Features and Outcomes from a Single-Center Experience

**DOI:** 10.3390/jcm14165614

**Published:** 2025-08-08

**Authors:** Radosław Dziedzic, Mariusz Korkosz, Joanna Kosałka-Węgiel

**Affiliations:** 1Jagiellonian University Medical College, Doctoral School of Medical and Health Sciences, św. Łazarza 16, 31-530 Kraków, Poland; radoslaw.dziedzic@doctoral.uj.edu.pl; 2Jagiellonian University Medical College, Department of Rheumatology and Immunology, Jakubowskiego 2, 30-688 Kraków, Poland; mariusz.korkosz@uj.edu.pl; 3University Hospital in Kraków, Department of Rheumatology, Immunology and Internal Medicine, Jakubowskiego 2, 30-688 Kraków, Poland

**Keywords:** diffuse alveolar hemorrhage, systemic lupus erythematosus, prognosis

## Abstract

**Background/Objectives**: Diffuse alveolar hemorrhage (DAH) is a rare but life-threatening complication that might occur in the course of systemic lupus erythematosus (SLE), presenting with acute respiratory symptoms, a rapid drop in hemoglobin, and diffuse pulmonary infiltrates. Despite various studies, clinical and laboratory risk factors for DAH in SLE remain unclear due to small cohort sizes and inconsistent findings. **Methods**: We analyzed the medical records of all adult SLE patients treated at the University Hospital in Kraków, Poland, from 2012 to 2022, to look for patients with DAH. **Results**: In a cohort of 1039 SLE patients, DAH was confirmed in five cases (0.48%), all presenting with respiratory symptoms and significant hemoglobin drops. No patients required intensive care unit admission or mechanical ventilation, and all survived the 5-year follow-up after receiving immunosuppressive therapy including glucocorticosteroids and cyclophosphamide, and also rituximab in one case. Common features included constitutional symptoms, hematologic and renal involvement, and frequent presence of antiphospholipid antibodies, with antiphospholipid syndrome diagnosed in three patients (60%). All patients had positive antinuclear antibodies, with the presence of anti-dsDNA and anti-SSA antibodies, each present in 3 out of 5 cases. **Conclusions**: In conclusion, early recognition and aggressive treatment of DAH in SLE patients, who often present other medical comorbidities as hematological, renal, and cardiovascular manifestations, is critical for improving long-term outcomes.

## 1. Introduction

Diffuse alveolar hemorrhage (DAH) remains one of the most life-threatening complications in patients with systemic lupus erythematosus (SLE), characterized by its low incidence but strikingly high mortality rate—reported to reach even up to 85.7% in some studies [1,2,3]. Clinically, DAH is defined as an acute syndrome marked by the sudden onset of non-specific respiratory symptoms such as hemoptysis, dyspnea, cough, sputum production, and hypoxia, along with a rapid drop in hemoglobin (Hb) levels and diffuse pulmonary infiltrates observed on chest imaging [1,3,4]. Radiologically and pathologically, it is characterized by transient alveolar infiltrates or consolidation, resulting from disruption of the alveolar–capillary basement membrane due to inflammatory damage or injury to small pulmonary vessels [4]. This leads to widespread extravasation of red blood cells into the alveolar spaces [4]. The precise pathogenesis of DAH in SLE remains incompletely understood [4].

Given the severity of this complication, timely differential diagnosis and prompt therapeutic intervention are of critical clinical importance to improve patient outcomes [1]. In light of this, a growing body of research has focused on identifying distinct clinical features and potential prognostic or mortality-related risk factors in SLE patients with DAH [1]. Several laboratory markers, including complement components (C3 and C4) and platelet counts, and clinical manifestations, such as neuropsychiatric involvement, serositis, pulmonary hypertension, and lupus nephritis (LN), have been reported to differ between SLE patients with and without DAH in selected case series [1,2]. However, these associations have not been consistent across all studies [1,2].

These discrepancies are likely influenced by the relatively small sample sizes in previous cohorts, which reflect the rarity of DAH in SLE and may limit the generalizability of findings [1,2,4]. Nonetheless, it remains imperative to better delineate the clinical and laboratory characteristics associated with DAH in this patient population. To contribute to this understanding, we conducted an analysis of medical records related to clinical and laboratory data of a relatively large single-center Polish cohort of five SLE patients with confirmed DAH, aiming to explore potential differentiating factors and identify patterns relevant to disease severity and prognosis.

## 2. Patients and Methods

### 2.1. Study Population

This study is based on a medical records analysis of hospital data of patients diagnosed with and treated for SLE at the University Hospital in Kraków, Poland, between 2012 and 2022. Only patients meeting the 2019 classification criteria established by the European League Against Rheumatism (EULAR) and American College of Rheumatology (ACR) were included [5].

We collected comprehensive data on patient demographics, including sex, age, and family history of autoimmune diseases, alongside clinical and laboratory findings. The data encompassed information on the onset of SLE symptoms, time to diagnosis, disease duration, associated comorbidities, history of miscarriages in women, treatment approaches, and—when applicable—causes and ages at death. Clinical manifestations assessed in this study included skin lesions, joint involvement, serositis, hematologic abnormalities, organ-specific complications (affecting the kidneys, liver, nervous system, and respiratory tract), Raynaud’s phenomenon, and lymphadenopathy. We have also added information regarding disease activity measurements during DAH, including SLEDAI-2K and/or BILAG. Detailed descriptions of medical comorbidities have been provided in our previous paper on SLE [6].

DAH was defined as the presence of at least three respiratory symptoms and signs, such as dyspnea, hypoxemia, hemoptysis, tachycardia, and/or cough, combined with diffuse interstitial and/or alveolar infiltrates observed on chest X-ray film or high-resolution computed tomography (HRCT), along with a sudden drop in hemoglobin level of at least 1.5 g/dL [3]. Additionally, the presence of hemosiderin-laden macrophages in bronchoalveolar lavage fluid (BALF) was recorded [3]. Cases due to other causes of alveolar bleeding—such as pulmonary thromboembolism, uremia, or acute pulmonary edema—were excluded, as were those not confirmed by BALF or endotracheal tube findings [7]. Categorical variables were expressed as counts and percentages. Continuous variables were presented as medians with Q1–Q3 ranges.

Ethical approval for this study was granted by the Bioethics Committee of Jagiellonian University Medical College (Approval No. 118.6120.21.2023, dated 15 June 2023). All research activities adhered to the ethical principles outlined in the Declaration of Helsinki.

### 2.2. Laboratory Analysis

Standard laboratory methods were employed to assess complete blood counts, lipid profiles, creatinine levels, and glomerular filtration rates (GFRs), which were calculated using the MDRD formula. Urine analyses included 24 h protein excretion and sediment examination. Antinuclear antibody (ANA) testing was conducted using indirect immunofluorescence on HEp-2 cells, followed by ELISA or line-blot assays to detect specific antibodies. Anti-dsDNA levels were measured using the *Crithidia luciliae* assay, while complement levels were determined via nephelometry. Tests for hypercoagulability included lupus anticoagulant screening and assessments of anti-cardiolipin and anti-β2 glycoprotein I antibodies (IgM and IgG).

## 3. Results

### 3.1. Demographic Features

DAH was confirmed in 5/1039 (0.48%) SLE patients from our SLE database: 3 women and 2 men. DAH was diagnosed at the time of SLE diagnosis confirmation in two cases (Patients No. 1 and No. 5). The median age at SLE onset was 29 years (range min–max: 26–62 years). Dyspnea, hemoptysis, or bleeding endotracheal tube occurred in all five SLE patients with DAH; cough, tachycardia, and hypoxia were observed in three, two, and one cases, respectively. New infiltrates on CXR or HRCT and a hemoglobin drop of at least 1.5 g/dL were evidenced in all SLE patients with DAH. Bronchoalveolar lavage was performed in two SLE patients, and hemosiderin-laden macrophages were observed in both. The median drop in hemoglobin (Hb) was 4.5 g/dL (2.2–5.1 g/dL). All analyzed patients presented a high SLE activity phenotype, defined as a SLEDAI-2K score ≥ 5 and the presence of at least one ‘A’ domain in the BILAG index. In patients No. 1 and No. 5, SLE was first diagnosed during the DAH complication. Detailed information has been provided in Table 1.

Regarding outcomes, none of the patients required hospitalization in the intensive care unit (ICU) or mechanical ventilation. None of the patients were concomitantly diagnosed with sepsis or macrophage activation syndrome (MAS). During the 5-year follow-up, none of the patients died. Intravenous methylprednisolone pulse therapy and intravenous cyclophosphamide were used in all SLE patients. One patient (No. 5) was additionally treated with intravenous rituximab. Detailed data are summarized in Table 2.

### 3.2. Clinical Manifestations

The most prevalent clinical manifestations were constitutional symptoms (*n* = 5, 100%), hematological abnormalities (*n* = 5, 100%), and kidney involvement (*n* = 4, 80%). A detailed analysis revealed that certain symptoms were present in all SLE patients diagnosed with DAH, specifically lymphopenia and anemia. Additionally, the vast majority of patients, four out of five (80%), exhibited fatigue/weakness, leukopenia, thrombocytopenia, leukocyturia, and 24 h urinary protein excretion exceeding 0.5 g/day.

Arthritis, arthralgia, pleural effusion, erythrocyturia, and interstitial lung disease (ILD) were observed in three out of five patients (60%). In two out of five cases (40%), symptoms such as fever, lymphadenopathy, pericardial effusion, urinary casts in the urine sediment, and 24 h urinary protein excretion greater than 3.5 g/day were noted.

Only one patient (20%) presented with lupus malar rash, alopecia, hemolytic anemia, and pulmonary hypertension. None of the patients exhibited myalgias, weight loss, discoid rash, urticaria, oral and/or nasal ulcers, Raynaud’s phenomenon, photosensitivity, pericarditis, or central or peripheral nervous system involvement. Detailed frequencies of all systemic symptoms across the SLE study groups are summarized in Table 3.

### 3.3. Comorbidities

The most common concomitant disease was arterial hypertension, diagnosed in four out of five SLE patients (80%), excluding patient No. 1. Heart failure was confirmed in three out of five patients (60%)—specifically, patients No. 2, No. 3, and No. 4. Similarly, hypercholesterolemia was identified in three out of five patients (60%): patients No. 2, No. 4, and No. 5. Deep venous thrombosis was diagnosed in two out of five patients (40%), namely patients No. 2 and No. 3. The following conditions were each diagnosed in only one patient: stroke in patient No. 1, transient ischemic attack in patient No. 2, and malignant tumor (breast cancer) in patient No. 4. None of the patients were diagnosed with hypothyroidism, hyperthyroidism, diabetes mellitus, atrial fibrillation, peripheral artery disease, end-stage kidney disease, myocardial infarction, or pulmonary embolism.

### 3.4. Autoantibody Profiles

All patients tested positive for antinuclear antibodies (ANAs). In all but one case, the ANA titer was 1:5120; patient No. 2 had an ANA titer of 1:2560 (the highest ANA titer ever stated). During the DAH occurrence, patient No. 1 presented an ANA titer of 1:1280, No. 2 presented a titer of 1:2560, and the rest of the patients (No. 3–5) had titers of 1:5120. In the follow-up analysis of 6–12 months after this severe complication, we had data on ANA titers in 3/5 analyzed patients (No. 1: 1:1280, No. 4: 1:640, and No. 5: 1:2560). Among SLE patients with DAH, the most frequently detected antibodies were anti-SSA (*n* = 3, 60%) and anti-dsDNA (*n* = 3, 60%). Specifically, anti-SSA antibodies were present in patients No. 1, No. 2, and No. 3, while anti-dsDNA antibodies were detected in patients No. 1, No. 3, and No. 5. The anti-dsDNA titers (CLIFT method) were 1:80 in patient No. 1, 1:2560 in patient No. 3, and 1:160 in patient No. 5. Anti-nucleosome antibodies were found in two cases (patients No. 1 and No. 3), and anti-RNP antibodies were also detected in two patients (patients No. 2 and No. 3). Additionally, patient No. 1 was positive for anti-SSB antibodies, and patient No. 5 was positive for anti-histone antibodies.

Two out of five patients were positive for anti-neutrophil cytoplasmic antibodies (ANCAs) without clinical or laboratory evidence of ANCA-associated vasculitis (AAV), as confirmed by ELISA: anti-proteinase 3 antibodies at 7.3 IU/mL (reference <2.0 IU/mL) in patient No. 3, and anti-myeloperoxidase antibodies at 5.3 IU/mL (reference <2.0 IU/mL) in patient No. 1.

Interestingly, all patients were positive for at least one antiphospholipid antibody (aPLA), with triple positivity observed in three patients (No. 1, No. 3, and No. 5). Patients No. 2 and No. 4 were positive only for anti-cardiolipin antibodies of the IgG and IgM classes. Antiphospholipid syndrome (APS) was diagnosed in patient No. 1 (stroke), patient No. 2 (transient ischemic attack), and patient No. 3 (deep venous thrombosis), based on the 2023 ACR/EULAR classification criteria. Detailed data are provided in Table 4.

### 3.5. Immunosuppressive Treatment

Glucocorticosteroids (GCS), mycophenolate mofetil (MMF), and cyclophosphamide (CTX) were administered in all five (100%) cases. Chloroquine or hydroxychloroquine (HCQ) was used in all but one case (except for patient No. 4). Additionally, patient No. 2 received treatment with azathioprine and cyclosporine A, while patient No. 5 was treated with rituximab and underwent plasmapheresis. No other immunosuppressive therapies, including methotrexate, sulfasalazine, belimumab, or anifrolumab, were administered to the analyzed patients during DAH presence.

Since SLE was first diagnosed in patients No. 1 and No. 5 during the episode of DAH, we have treatment history only for the remaining patients: patient No. 2 had previously received GCS and CTX; patient No. 3 had been treated with HCQ, GCS, and MMF; and patient No. 5 had a history of treatment with HCQ, GCS, CTX, and MMF.

## 4. Discussion

Our research provides insights into the clinical and laboratory profiles of a cohort of SLE patients from a single center, with a particular focus on those with DAH. Our findings reveal that DAH alters the clinical course of the disease. To our knowledge, this is one of the largest studies of Polish SLE patients to date, analyzing various aspects of the disease in relation to the presence of DAH. DAH is a rare manifestation of SLE [7]. In our cohort, only 0.48% of SLE patients exhibited this complication. These findings are consistent with studies on larger populations, where DAH occurrence ranged from 0.5% to 9% across cohorts of 8 to 29 patients [7,8,9,10,11,12,13,14,15]. Interestingly, this variability in reported DAH frequency may result from differences in diagnostic criteria applied for this SLE manifestation. Therefore, these results should be interpreted with caution [3]. In our study, we ultimately included only five DAH cases. An additional four patients were excluded from the DAH subgroup despite an initial DAH diagnosis in their medical records, as they did not meet the inclusion criteria.

Our demographic analysis showed that there were three women and two men in our cohort. Most studies report that DAH occurs more frequently in women; however, it is important to consider that SLE itself predominantly affects women, with approximately 90% of all cases occurring in the female population [7,16]. Therefore, the observed high frequency of DAH in women may, at least in part, be relative and reflect the underlying sex distribution of SLE rather than a true increased risk specific to female patients. Next, most case series have reported that DAH tends to occur early in the course of SLE [7]. Similarly, in our study, the median time from SLE onset to DAH occurrence was three years. In other studies, the mean or median time to DAH onset ranged from 6 months to 14.1 years [7,17,18].

In our cohort, none of the patients died. Our results are consistent with the data presented by Santos-Ocampo et al. [14], who reported 11 cases of DAH, all of whom survived. However, in most studies, the mortality risk associated with DAH in SLE was significantly higher, reaching over 90% of cases in the study by Abud-Mendoza et al. [19]. These differences in survival may result from the fact that the analyses were conducted over a period of approximately 40 years, which likely influenced diagnostic and therapeutic approaches [7]. Additionally, the outcomes might vary due to the severity of bleeding into the respiratory tract, as indicated by a decrease in Hb levels ranging from 1.9 to 5.5 g/dL [7,9,20]. Nevertheless, despite the relatively high median Hb drop (4.5 g/dL) in our cases, all patients survived.

The most frequently observed clinical manifestations in our cohort were general (constitutional) symptoms, which were present in all patients, hematological abnormalities, and kidney involvement, each diagnosed in all but one case. Similar to our findings, the majority of studies report kidney involvement in at least two-thirds of SLE patients with DAH [7,10,15,19]. In our cohort, the most common renal manifestation was proteinuria. Data regarding concomitant arthritis are inconsistent across studies, with reported prevalence ranging from 10% to 75.9%. In our study, arthritis was present in 60% of cases, which places our results within the middle range of previously reported data. Other severe SLE manifestations, particularly central nervous system involvement, were not observed in our cohort. However, the rarity of central nervous system involvement in DAH cases has also been emphasized by other authors [9,10,11,12,13,14,15,17,18,20,21,22].

Interestingly, in our cohort, arterial hypertension was confirmed in four cases, while heart failure and hypercholesterolemia (each diagnosed in three cases) were the most frequently observed comorbidities. It is important to emphasize that uncontrolled arterial hypertension and heart failure may contribute to or exacerbate DAH [23,24]. Therefore, it is crucial to manage not only SLE but also internal comorbidities. Avoiding fluid overload and hyperdynamic circulation may play a key role in improving outcomes for patients with DAH [7,23,24]

SLE is a complex, immune-mediated disease characterized by the presence of a wide variety of autoantibodies with different clinical patterns [25,26]. In our study, the antibody profiles of SLE patients with DAH indicated that certain ANAs were most frequently present, specifically anti-SSA and anti-dsDNA antibodies, both confirmed in three cases. The higher prevalence of anti-dsDNA antibodies may correlate with the high incidence of LN in our cohort, further supporting the established association between these autoantibodies and kidney involvement [27]. Similarly, Kwok et al. [11] also emphasized that anti-SSA and anti-dsDNA are the most frequently observed antinuclear autoantibodies in SLE patients with DAH. Interestingly, in two cases (No. 1 and No. 3, accounting for 40% of the analyzed cohort), ANCAs were confirmed using the ELISA method. This finding suggests that ANCAs may be more commonly detected in SLE patients with DAH, which is consistent with previous reports indicating ANCA positivity in up to 9.3% of all SLE cases [28]. However, due to the small sample size in our study, further analysis on a larger cohort of SLE patients with DAH is necessary to validate these observations.

Regarding aPLA, they were present in all cases in our study, with triple-positive APS confirmed in three patients (60%). Additionally, three patients experienced thromboembolic events. Our findings contrast with those of Kwok et al. [11], who reported the presence of aPLA in only a minority of DAH cases, with no significant difference compared to SLE patients without DAH. It should be considered that DAH may also be a complication of APS, highlighting the need for careful monitoring of this subset of SLE patients for potential DAH development [29].

Regarding treatment, SLE patients with DAH in our cohort were more frequently prescribed aggressive immunosuppressive therapies, including systemic GCS (sGCS), MMF, and CTX, all of which were administered in every case. This reflects the more severe nature of their disease, as these treatments are typically reserved for patients with active or refractory disease [30]. The fact that the DAH group required such intensive immunosuppressive regimens underscores the importance of individualized treatment strategies for patients with this serious complication. Interestingly, sGCS and CTX were not universally used in all DAH cases reported in other studies [7]. Given the severity of the disease in our cohort, one patient (No. 5) was additionally treated with rituximab and underwent plasmapheresis. Compared to other studies, there is significant variability in the reported frequency of plasmapheresis use among patients with DAH [8,12,13,17,18,20,21,22,31,32]. Next, intensive immunosuppressive treatment appears to be associated with favorable outcomes in SLE-related DAH, which aligns with recent advances in therapeutic approaches. Rituximab (RTX), recommended in updated guidelines for refractory or severe SLE, may represent a promising treatment option in selected DAH cases [33]. In our study, RTX was used in addition to GCS and CTX in patient No. 5.

We would like to acknowledge several limitations of our study. First, its design (a medical records review) introduces potential biases related to both data collection and patient selection. Additionally, the number of confirmed SLE patients with DAH is small (*n* = 5/1039 [0.48%]). Thus, for this reason, we did not perform a comparative analysis with SLE patients without DAH. Moreover, the paper presents a single-center experience with DAH from the rheumatology department in Kraków, Poland. However, considering the rarity of this complication, we believe the results remain valuable. The single-center nature of the study may also limit the generalizability of our findings to broader or more diverse populations. Another limitation is the lack of patient-reported outcomes, such as quality-of-life assessments, which could have provided a more comprehensive understanding of patient well-being and the broader impact of both the disease and its treatment. Finally, some of the observed associations may be incidental rather than indicative of true causality.

## 5. Conclusions

In conclusion, DAH in SLE patients is linked to a higher risk of hematological issues, lupus nephritis, and comorbidities such as hypertension, heart failure, and antiphospholipid syndrome. Due to its severity, early recognition and aggressive treatment are vital to improving outcomes and reducing mortality.

## Figures and Tables

**Table 1 jcm-14-05614-t001:** Demographic, clinical manifestations, and imaging in five systemic lupus erythematosus patients with diffuse alveolar hemorrhage form our study cohort.

No. of the DAH Patient	No. 1	No. 2	No. 3	No. 4	No. 5
**Sex (female/male)**	Female	Male	Male	Female	Female
**Age at the SLE onset, years**	32	26	29	28	62
**Time from the SLE onset to DAH diagnosis, years**	0	3	6	32	0
**Cough (presence)**	Y	Y	Y	N	N
**Dyspnea (presence)**	Y	Y	Y	Y	Y
**Tachycardia (presence)**	N	N	Y	N	N
**Hypoxia (presence)**	N	N	N	Y	Y
**Hemoptysis or bleeding in endotracheal tube**	Y	Y	Y	Y	Y
**New infiltrates CXR or HRCT (presence)**	Y	Y	Y	Y	Y
**Drop in hemoglobin levels, g/dL**	1.5	2.2	5.6	4.5	5.1
**SLEDAI-2K, points**	5	11	11	17	16
**BILAG index**	1A2B	2A1C	2A2B1C	2A1B	3A1B1C

Abbreviations: BILAG—British Isles Lupus Assessment Group Index; CXR—chest X-ray film; DAH—diffuse alveolar hemorrhage; HRCT—high-resolution computed tomography; N—no; SLE—systemic lupus erythematosus; SLEDAI-2K—Systemic Lupus Erythematosus Disease Activity Index 2000; Y—yes.

**Table 2 jcm-14-05614-t002:** Outcomes and immunosuppressive treatment in systemic lupus erythematosus patients with diffuse alveolar hemorrhage.

No. of the DAH Patient	No. 1	No. 2	No. 3	No. 4	No. 5
**ICU**	N	N	N	N	N
**Mechanical ventilation**	N	N	N	N	N
**Sepsis**	N	N	N	N	N
**MAS**	N	N	N	N	N
**Death during the DAH course**	N	N	N	N	N
**IV methylprednisolone pulse therapy**	Y	Y	Y	Y	Y
**IV cyclophosphamide**	Y	Y	Y	Y	Y
**IV rituximab**	N	N	N	N	Y

Abbreviations: DAH—diffuse alveolar hemorrhage; ICU—intensive care unit; IV—intravenous; MAS—macrophage activation syndrome; N—no; Y—yes.

**Table 3 jcm-14-05614-t003:** Presence of systemic involvement in the systemic lupus erythematosus cohort with diffuse alveolar hemorrhage.

Clinical Manifestations	SLE Patients with DAH *n* = 5
Constitutional symptoms, *n* (%)	5 (100%)
Fever, *n* (%)	2 (40%)
Fatigue/weakness, *n* (%)	4 (80%)
Myalgias, *n* (%)	0 (0%)
Weight loss, *n* (%)	0 (0%)
Lymphadenopathy, *n* (%)	2 (40%)
Mucocutaneous manifestations, *n* (%)	1 (20%)
Lupus malar rash, *n* (%)	1 (20%)
Discoid rash, *n* (%)	0 (0%)
Urticaria, *n* (%)	0 (0%)
Alopecia, *n* (%)	1 (20%)
Oral and/or nasal ulcers, *n* (%)	0 (0%)
Photosensitivity, *n* (%)	0 (0%)
Joint manifestations, *n* (%)	3 (60%)
Arthritis, *n* (%)	3 (60%)
Arthralgia, *n* (%)	3 (60%)
Serositis, *n* (%)	3 (60%)
Pleural effusion, *n* (%)	3 (60%)
Pericardial effusion, *n* (%)	2 (40%)
Pericarditis, *n* (%)	0 (0%)
Hematological manifestations, *n* (%)	5 (100.0%)
Leucopenia ^#^, *n* (%)	4 (80%)
Lymphopenia ^$^, *n* (%)	5 (100.0%)
Anemia ^&^, *n* (%)	5 (100.0%)
Hemolytic anemia *, *n* (%)	1 (20%)
Thrombocytopenia ^@^, *n* (%)	4 (80%)
Kidney involvement, *n* (%)	4 (80%)
24 h urinary protein excretion > 0.5 g/day, *n* (%)	4 (80%)
24 h urinary protein excretion > 3.5 g/day, *n* (%)	2 (40%)
Urinary casts, *n* (%)	2 (40%)
Erythrocyturia, *n* (%)	3 (60%)
Leucocyturia, *n* (%)	4 (80%)
Neurological signs, *n* (%)	0 (0%)
Central nervous system involvement, *n* (%)	0 (0%)
Peripheral nervous system involvement, *n* (%)	0 (0%)
Raynaud’s phenomenon, *n* (%)	0 (0%)
Lung involvement, *n* (%)	3 (60%)
Interstitial lung disease, *n* (%)	3 (60%)
Hemolytic anemia, *n* (%)	1 (20%)
Autoimmune hepatitis, *n* (%)	0 (0%)

Categorical variables are presented as numbers with percentages. Abbreviations: *n*—number; SLE—systemic lupus erythematosus; DAH—diffuse alveolar hemorrhage. ^#^—<4000/µL or diagnosis based on medical history; ^$^—<1500/µL or diagnosis based on medical history; ^&^—≤12 g/dL in women; ≤13.5 g/dL in men; or diagnosis based on medical history; *—anemia with positive direct Coombs test; anemia with decreased haptoglobin level; or diagnosis based on medical history; ^@^—<100,000/µL or diagnosis based on medical history.

**Table 4 jcm-14-05614-t004:** Antibody profiles in systemic lupus erythematosus patients with diffuse alveolar hemorrhage.

Parameter	SLE Patients with DAH *n* = 5
Anti-SSA antibodies, *n* (%)	3 (60%)
Anti-SSB antibodies, *n* (%)	1 (20%)
Anti-histone antibodies, *n* (%)	1 (20%)
Anti-nucleosome antibodies, *n* (%)	2 (40%)
Anti-Smith antibodies, *n* (%)	0 (0%)
Anti-RNP antibodies, *n* (%)	2 (40%)
Anti-dsDNA antibodies, *n* (%)	3 (60%)
Lupus anticoagulant, *n* (%)	3 (60%)
Anti-cardiolipin IgG and/or IgM, *n* (%)	5 (100%)
Anti-cardiolipin IgG, GPL	45.53 (37.26–60.77)
Anti-cardiolipin IgM, MPL	0 (0–19.05)
Anti-β2 glycoprotein I IgG and/or IgM, *n* (%)	3 (60%)
Anti-β2 glycoprotein I IgG, SGU	0 (0–124.6)
Anti-β2 glycoprotein I IgM, SMU	18.3 (0–70.48)
At least one positive antiphospholipid antibody	5 (100%)
Triple-positive antiphospholipid antibodies	3 (60%)
Antiphospholipid syndrome ^#^, *n* (%)	3 (60%)

Categorical variables are presented as numbers with percentages. Abbreviation: DAH—diffuse alveolar hemorrhage; *n*—number; SLE—systemic lupus erythematosus. ^#^—according to the 2023 ACR/EULAR classification criteria.

## Data Availability

The data presented in this study are available on reasonable request from the corresponding author.

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
