# Peer review of "Characteristics of Systemic Lupus Erythematosus Patients with Diffuse Alveolar Hemorrhage: Clinical Features and Outcomes from a Single-Center Experience"

_jcm, 2025, doi:10.3390/jcm14165614_

Round 1
Reviewer 1 Report
Comments and Suggestions for Authors
Characteristics of SLE patients with DAH, although the sample was rather small(only 5 patients), it is meaningful for early recognition and aggressive treatment of DAH in SLE patients.
1.The disease activity and organ damage were also very important to illustrate. These SLE patients were recommended to add SLEDAI and BILAG.
2.The medications such as GCs, immunosuppressives and biologics were also the confounding factors, which also need to be added.
Author Response
Manuscript ID: jcm-3767953
Title: “Characteristics of systemic lupus erythematosus patients with diffuse alveolar hemorrhage: clinical features and outcomes from a single-center retrospective study”
Reviewer 1
Characteristics of SLE patients with DAH, although the sample was rather small(only 5 patients), it is meaningful for early recognition and aggressive treatment of DAH in SLE patients.
General response to comments:
The authors sincerely appreciate Reviewer 1 for the thorough evaluation of our work and for offering insightful suggestions that have enhanced the quality of our manuscript. Below, we have provided point-by-point responses to the comments.
1.The disease activity and organ damage were also very important to illustrate. These SLE patients were recommended to add SLEDAI and BILAG.
Response:
Thank you for this valuable suggestion. We noticed coexistence of several comorbidities in SLE patients with DAH, thus, we believe that presenting them in a summarized table is valuable for our report. Thank you for understanding. We have also added SLE-related disease activity measurements such as SLEDAI-2K and BILAG during the course of DAH in all included patients. We have added this information to the Methods section.
2.The medications such as GCs, immunosuppressives and biologics were also the confounding factors, which also need to be added.
Response:
Thank you for this valuable comment. We have presented the treatment strategies of the DAH patients included in the analysis in paragraph 3.5 of the Results section, in order to show the medications used in the illustrated cases, including those administered prior to the diagnosis of SLE. Additionally, information on immunosuppressive treatment during the DAH episode has been provided in Table 2.
We hope that the current version of the manuscript is suitable for publication. Once again, thank you very much for your valuable comments and suggestions.
Reviewer 2 Report
Comments and Suggestions for Authors
This is an interesting report of a relatively rare event occurring in patients with systemic lupus erythematosus (SLE). Data omitted detract, however, from the value of this communication, specifically, patients' disease activity and damage accrual using standard and validated instruments have not been included; such data would have allowed to better understand the characteristics of these SLE patients. In addition, please note the following:
*To better understand the value of this communication, why were patients with and those without DAH not compared?
*This registry, sensu strictu, this is not a retrospective study if we use the correct epidemiological definition of this term: that is from the disease Y/N to the attribute Y/N, like in Lung Cancer Y/N and Smoking Y/N; rather this is a medical records review or historical study.
*It is not Chest X-ray bu Chest radiograph or Chest X-ray film.
*Table 3 is cluttered (repeating % signs in the body of the table); it also includes terminology no longer in use such as lupoid hepatitis. Moreover, what is the purpose of including clinical items for which no patients were affected by them? In addition, no definitions are provided for these and other items in this table
*The headings in Tables 1 and 2 should be listed horizontally rather than vertically to make their reading easier.
*Finally, a much shorter communication is in order.
Comments on the Quality of English Language
Overall, it could be improved.
Author Response
Manuscript ID: jcm-3767953
Title: “Characteristics of systemic lupus erythematosus patients with diffuse alveolar hemorrhage: clinical features and outcomes from a single-center retrospective study”
Reviewer 2
This is an interesting report of a relatively rare event occurring in patients with systemic lupus erythematosus (SLE). Data omitted detract, however, from the value of this communication, specifically, patients' disease activity and damage accrual using standard and validated instruments have not been included; such data would have allowed to better understand the characteristics of these SLE patients. In addition, please note the following:
General response to comments:
The authors would like to thank Reviewer 2 for the thorough evaluation of our work and for the insightful comments that helped improve the manuscript. In the revised version, we have added information on lupus disease activity assessments, including the SLE Disease Activity Index 2000 (SLEDAI-2K) and the British Isles Lupus Assessment Group index (BILAG), during the occurrence of diffuse alveolar hemorrhage (DAH). This information is presented in Table 1. Below, we provide point-by-point responses to the reviewer’s comments.
*To better understand the value of this communication, why were patients with and those without DAH not compared?
Response:
Thank you very much for the comment. The primary aim of our article was to provide a detailed characterization of patients with SLE who developed a very rare complication — diffuse alveolar hemorrhage (DAH). In our hospital database, we identified 1,039 patients with SLE, among whom only 5 developed DAH. We believe that comparing SLE patients with DAH to those without this complication could introduce bias due to the extreme rarity of DAH. Therefore, although we initially considered such a comparative analysis, we ultimately chose to focus on an in-depth description of the DAH cases in order to provide clinically meaningful data and to avoid biased results from comparing 5 patients to a cohort of 1,034. This has now been acknowledged in the limitations section of the study.
*This registry, sensu strictu, this is not a retrospective study if we use the correct epidemiological definition of this term: that is from the disease Y/N to the attribute Y/N, like in Lung Cancer Y/N and Smoking Y/N; rather this is a medical records review or historical study.
Response:
Thank you very much for the comment. We agree about the distinction in the definition of a retrospective study from an epidemiological standpoint. In fact, that this registry does not constitute a retrospective study in the strict epidemiological sense, as it does not follow the direction from disease to exposure. Therefore, referring to it as a medical records review or historical cohort study would be more precise. We have revised the terminology accordingly to improve clarity and accuracy to: a medical records review.
*It is not Chest X-ray bu Chest radiograph or Chest X-ray film.
Response:
Thank you very much for your comment. We have revised it accordingly.
*Table 3 is cluttered (repeating % signs in the body of the table); it also includes terminology no longer in use such as lupoid hepatitis. Moreover, what is the purpose of including clinical items for which no patients were affected by them? In addition, no definitions are provided for these and other items in this table
Response:
Thank you very much for your comment. We have corrected the table according to your suggestion. We have changed the name for lupoid hepatits to autoimmune hepatitis. Next, we added dull descriptions of medical comorbidities in our previous paper (doi: 10.1007/s00296-024-05579-4).
*The headings in Tables 1 and 2 should be listed horizontally rather than vertically to make their reading easier.
Response:
Thank you very much for this comment. We have revised it and now Tables 1 and 2 are listed horizontally as the rest part of the manuscript. We believe that in this way now it easier for potential readers to follow.
*Finally, a much shorter communication is in order.
Response:
Thank you very much for this comment. We agree with your suggestion. We have revised the Conclusions section to enhance the clarity and readability.
We hope that the current version of the manuscript is suitable for publication. Once again, thank you very much for your valuable comments and suggestions.
Reviewer 3 Report
Comments and Suggestions for Authors
The manuscript titled “Characteristics of systemic lupus erythematosus patients with diffuse alveolar hemorrhage: clinical features and outcomes from a single-center retrospective study” is well written and presented.
This study highlights that SLE patients with DAH are at higher risk for hematological issues, lupus nephritis, and comorbidities like hypertension and APS. DAH is a severe complication requiring early detection and aggressive treatment. Prompt management is key to improving survival and long-term outcomes.
This manuscript needs a minor revision to improve clarity and incorporate specific feedback:
- This study contains only 5 confirmed DAH cases out of 1,039 SLE patients (~0.48%) were analyzed, which is too small to draw statistically significant or generalizable conclusions.
- There is no comparison group (e.g., SLE patients without DAH) is included in this study, which hinders evaluation of specific DAH risk factors or predictors.
- The study spans between 2012 to 2022, a decade during which treatment standards and diagnostics may have evolved, introducing variability.
- The study is only conducted at one hospital in Kraków, Poland, limiting external validity and applicability to broader, more diverse populations.
- Although serological findings are mentioned (e.g., ANA, anti-dsDNA, anti-SSA), their clinical significance in DAH prediction or severity is not analyzed in depth.
- The authors are encouraged to elaborate more details on the clinical significance and wider implications of this study.
- Few sentences could be rephrased for better clarity and readability.
Line 68-69: "Only patients meeting the 2019 classification criteria established by EULAR and ACR were included [5]."
Line 104: "SLE patients" is repeated.
Line 106: "The median age at SLE onset was 29 years (range: 28–32 years)."
Line 110: macrophages were observed in both.
Line 113: "no of the patients" → "none of the patients"
Line 115: "During the 14-year follow-up, none of the patients died."
Author Response
Manuscript ID: jcm-3767953
Title: “Characteristics of systemic lupus erythematosus patients with diffuse alveolar hemorrhage: clinical features and outcomes from a single-center retrospective study”
Reviewer 3
The manuscript titled “Characteristics of systemic lupus erythematosus patients with diffuse alveolar hemorrhage: clinical features and outcomes from a single-center retrospective study” is well written and presented.
This study highlights that SLE patients with DAH are at higher risk for hematological issues, lupus nephritis, and comorbidities like hypertension and APS. DAH is a severe complication requiring early detection and aggressive treatment. Prompt management is key to improving survival and long-term outcomes.
General response to comments:
The authors sincerely appreciate Reviewer 3 for the thorough evaluation of our work and for offering insightful suggestions that have enhanced the quality of our manuscript. Below, we have provided point-by-point responses to the comments.
This manuscript needs a minor revision to improve clarity and incorporate specific feedback:
- This study contains only 5 confirmed DAH cases out of 1,039 SLE patients (~0.48%) were analyzed, which is too small to draw statistically significant or generalizable conclusions.
Response:
Thank you very much for your comment. We agree that our analysis includes a relatively small sample of patients with DAH. However, this is a single-center study based on more than a decade of observation of SLE patients. Although this complication is rare, we believe our report offers a unique insight into this manifestation of SLE. Nevertheless, we have acknowledged this limitation in the Limitations section at the end of the Discussion.
- There is no comparison group (e.g., SLE patients without DAH) is included in this study, which hinders evaluation of specific DAH risk factors or predictors.
Response:
Thank you very much for the comment. The main aim of our article was to thoroughly characterize patients with SLE who developed a very rare complication — diffuse alveolar hemorrhage (DAH). In our hospital database, we identified 1,039 patients with SLE, of whom only 5 developed DAH. We believe that comparing DAH-SLE patients with non-DAH SLE patients would lead to biased results due to the extreme rarity of DAH. Therefore, although we initially considered such an analysis, we ultimately decided to focus on an in-depth characterization of the DAH cases in order to provide clinically valuable data. We have added this issue to the Limitation of the study.
- The study spans between 2012 to 2022, a decade during which treatment standards and diagnostics may have evolved, introducing variability.
Response:
Thank you very much for this valuable comment. We agree that the time of diagnosis could have influenced therapeutic decisions, particularly due to the evolving recommendations for the management of SLE, such as the updated 2019 EULAR guidelines. For example, one of the patients was diagnosed in 2021 and received also rituximab (RTX) as part of a more intensive immunosuppressive regimen, which reflects changes in clinical practice and the availability of newer therapies. In response to your suggestion, we have now included this temporal context in the Discussion section and clarified the differences in immunosuppressive treatment related to the year of diagnosis and DAH onset. Please see updated immunosuppressive treatment data for our DAH cases.
- The study is only conducted at one hospital in Kraków, Poland, limiting external validity and applicability to broader, more diverse populations.
Response:
Thank you very much for your comment. We are aware that this is a single-center analysis; however, we analyzed data from over 1,000 patients with SLE, which represents a substantial cohort. Given the limited number of publications on DAH in SLE and the typically small sample sizes in those studies, we hope that our findings will contribute meaningful insights. Nonetheless, we acknowledge the need for future multicenter studies to validate these results and support broader clinical applicability. This limitation has been added to the Limitations section in the end of the paper.
- Although serological findings are mentioned (e.g., ANA, anti-dsDNA, anti-SSA), their clinical significance in DAH prediction or severity is not analyzed in depth.
Response:
Thank you very much for this insightful comment. We have expanded the text to include more detailed information on antinuclear antibodies (ANA) in the 3.4 paragraph, including the highest ANA titers ever recorded as well as those present at the time of DAH onset. Additionally, we now provide data on specific autoantibodies: each anti-dsDNA and anti-Ro (SSA) antibodies were detected in 3 out of 5 patients during the DAH episode, making them the most commonly observed ANA specificities in our cohort.
- The authors are encouraged to elaborate more details on the clinical significance and wider implications of this study.
Response:
Thank you very much for the comment. We have expanded the Conclusions section to better highlight the clinical significance and broader implications of our findings. The effectiveness of intensive immunosuppressive therapy in treating life-threatening DAH in SLE patients is consistent with evidence from previous large reviews, supporting the need for early and aggressive management in such cases. Our observations emphasize that certain high-risk patients, such as those with coexisting lupus nephritis, heart failure, or antiphospholipid syndrome, require particular diagnostic vigilance. These comorbidities may increase the likelihood of DAH or complicate its course, reinforcing the importance of tailored clinical monitoring and intervention strategies in this vulnerable subgroup.
- Few sentences could be rephrased for better clarity and readability.
Line 68-69: "Only patients meeting the 2019 classification criteria established by EULAR and ACR were included [5]."
Line 104: "SLE patients" is repeated.
Line 106: "The median age at SLE onset was 29 years (range: 28–32 years)."
Line 110: macrophages were observed in both.
Line 113: "no of the patients" → "none of the patients"
Line 115: "During the 14-year follow-up, none of the patients died."
Response:
Thank you very much for that comment. We have revised it accordingly.
We hope that the current version of the manuscript is suitable for publication. Once again, thank you very much for your valuable comments and suggestions.
Round 2
Reviewer 2 Report
Comments and Suggestions for Authors
The authors have addressed my previous concerns. No further issues.